# Association between Irregular Meal Timing and the Mental Health of Japanese Workers

**DOI:** 10.3390/nu13082775

**Published:** 2021-08-13

**Authors:** Yu Tahara, Saneyuki Makino, Takahiko Suiko, Yuki Nagamori, Takao Iwai, Megumi Aono, Shigenobu Shibata

**Affiliations:** 1Laboratory of Physiology and Pharmacology, School of Advanced Science and Engineering, Waseda University, Shinjuku-ku, Tokyo 162-8480, Japan; s.makino@fuji.waseda.jp; 2Research and Development Headquarters, Lion Corporation, Edogawa, Tokyo 132-0035, Japan; t-suiko@lion.co.jp (T.S.); yk-ngmr@lion.co.jp (Y.N.); i-takao@lion.co.jp (T.I.); meg-y@lion.co.jp (M.A.)

**Keywords:** chrono-nutrition, circadian clock, sleep

## Abstract

Breakfast skipping and nighttime snacking have been identified as risk factors for obesity, diabetes, and cardiovascular diseases. However, the effects of irregularity of meal timing on health and daily quality of life are still unclear. In this study, a web-based self-administered questionnaire survey was conducted involving 4490 workers (73.3% males; average age = 47.4 ± 0.1 years) in Japan to investigate the association between meal habits, health, and social relationships. This study identified that irregular meal timing was correlated with higher neuroticism (one of the Big Five personality traits), lower physical activity levels, and higher productivity loss. Irregular meal timing was also associated with a higher incidence of sleep problems and lower subjective health conditions. Among health outcomes, a high correlation of irregular meal timing with mental health factors was observed. This study showed that irregularity of meal timing can be explained by unbalanced diets, frequent breakfast skipping, increased snacking frequency, and insufficient latency from the last meal to sleep onset. Finally, logistic regression analysis was conducted, and a significant contribution of meal timing irregularity to subjective mental health was found under adjustment for other confounding factors. These results suggest that irregular meal timing is a good marker of subjective mental health issues.

## 1. Introduction

Health management of employees in the workplace is shown to have improved their performance (that is, leading to a reduction in absenteeism and an increase in presenteeism) as well as overall corporate performance. This strategy is now receiving attention worldwide [1,2,3]. However, the protocol for health management is under development and needs to be verified with scientific evidence [4]. Daily health management includes the control of daily behaviors, such as physical activity, food, and sleep [5,6].

Eating habits and nutrition are critically associated with mood, mental wellbeing, and sleep [7]. Western dietary habits not only increase the risk of obesity but also pose a risk to mental health [8,9]. A higher intake of fruits, vegetables, and proteins is characterized as good nutrition for maintaining health. These food groups include the following: eicosapentaenoic acid (EPA), docosahexaenoic acid (DHA), alpha-tocopherol, magnesium, and folic acid, which have been reported as functional nutrients for mental health [8,9]. Additionally, polyphenol compounds are good for mental health: they modulate brain function, gut microbiota, and oxidative stress/inflammation signaling [10]. In addition to specific food groups, eating behavior is also an important factor in this regard. Breakfast skipping has been associated with higher levels of stress, depression, and fatigue [11,12]. Patients with night eating syndrome (NES) show nocturnal emotional eating with a higher depressed mood in the evening and lower sleep quality [13,14].

Irregular sleep/wake patterns with disrupted circadian rhythms directly affect mental wellbeing [15]. The circadian clock is a homeostatic regulation system consisting of clock genes in the body, which maintain physiological functions in line with a 24 h cycle [16]. Chronic jet lag in night shift work has been identified as a risk factor for sleep disorders, depression, obesity, diabetes, and cancer [17]. The circadian clock can be adjusted with proper light, food, and exercise [18]. Therefore, the timing of these environmental factors must be considered in order to maintain the regularity of the circadian clock. Research on the timing, quality, and quantity of food intake, known as “chrono-nutrition”, is a growing field [19]. It is important to note that breakfast skipping and nighttime eating delay the circadian clock and lead to obesity [20]. On the other hand, maintaining regular meal timing and following strategies such as time-restricted eating (TRE), in which the time from the first meal to the last meal in a day is restricted to a period of 8–12 h, have been reported to improve circadian clock oscillation, obesity, pre-type 2 diabetes, and cognitive function [21,22]. TRE has also been reported to be beneficial for aging, neurodegenerative disorders, and cancer [21]. However, data on the effect of irregular meal timing on mental health remain elusive.

In the current study, we evaluated the association between eating habits, subjective health outcomes, and social relationships in Japanese workers using a web-based self-administered questionnaire survey. Although self-administered surveys have limitations, the results of the analysis of the survey responses strongly suggest that irregular meal timing is directly associated with subjective mental health.

## 2. Materials and Methods

### 2.1. Ethical Approval

The Ethics Review Committee on Research with Human Subjects at Waseda University and Lion Corporation approved this experiment (No. 2020-046 and No. 349, respectively), and the guidelines laid down in the Declaration of Helsinki were followed.

### 2.2. Target Population and Data Collection

Participants were recruited through an online survey company (Cross Marketing Inc., Tokyo, Japan) from 19 December 2020 to 25 December 2020. Japanese male and female workers, aged 20 to 69 years old and living in Japan (from Hokkaido to Okinawa), were asked to participate in this web survey. In total, 8720 participants responded to the questionnaire. Since the survey was conducted using an online web system, participants who were not working at the time, who were under 20 years old or older than 69 years old, or who did not complete the questionnaire appropriately were automatically excluded. Thus, out of a total of 8720 participants, 4490 participants (73.3% males; 47.4 ± 0.1 y) ultimately completed the questionnaire. The sex ratio of the participants was similar to the sex ratio of Japanese full-time workers as reported in the Japanese national survey from Statistics Bureau (https://www.stat.go.jp/english/data/index.html; access date: 1 June 2021).

### 2.3. Variables

The questionnaire included 276–318 questions depending on work type (e.g., shift work or otherwise); the participants answered them within 30–50 min. In the current study, we focused on the below-described variables from the questionnaire. The basic characteristics of the participants included age, sex, body mass index (BMI), and whether or not they were night shift workers. Subjective wellbeing was evaluated using the Satisfaction with Life Scale (SWLS) [23]. The subjects’ personalities were assessed using the Big Five personality traits [24]. Their daily physical activity was evaluated using the short version of the International Physical Activity Questionnaire (IPAQ) [25]. The subjects’ performances in the working environment were investigated using the short Japanese version of the WHO Health and Work Performance Questionnaire (WHO-HPQ) [26,27]. Chronotype was evaluated from the mid-point of the sleep phase on free days using the Munich ChronoType Questionnaire (MCTQ) [28]. Sleep quality and sleep problems were also evaluated using a seven-step selection questionnaire (1 = strongly disagree, 2 = disagree, 3 = slightly disagree, 4 = neither agree or disagree, 5 = slightly agree, 6 = agree, 7 = strongly agree), and the questions were designed with reference to the Japanese version of the Sleep Quality Index (PSQI-J) [29]. Subjective health and mental health were also assessed using the seven-step selection questionnaire. The subjective mental health questionnaire was developed with reference to the Japanese Stress Check Program, which is currently authorized by the Japanese Ministry of Health, Labor and Welfare and is considered a standard measure for the assessment of occupational stress [30,31]. We selected 10 categories from the Japanese Stress Check Program to assess the subjects’ stress, anxiety, and depression levels. The instruction “Think about your mental health condition: I feel very tired”, along with the seven-step selection answer scale as described above, was used.

Daily eating habits were assessed using a seven-step selection questionnaire. The questionnaire assessed whether the subjects had engaged in the following behaviors in the past 4 weeks: frequent irregular timing of daily food intake, lower number of chewing cycles, shorter eating duration at each meal, eating a large amount at each mealtime, eating out frequently, eating unbalanced diets frequently, eating salty food frequently, and low vegetable intake. For example, the instruction used to assess the irregularity of meal timing was “Think about your eating behavior in the past 4 weeks: meal timing is irregular”, followed by the seven-step answer scale as described above. The ratio of the size of each meal (breakfast, lunch, dinner, and snack; total = 10) was also determined.

### 2.4. Statistical Analysis

Statistical analysis was conducted using SPSS (version 27, IBM, IL, USA). Correlation analysis was performed using Spearman’s rank correlation test. Non-parametric analysis (Kruskal–Wallis test) was also applied to analyze the eating habits of the subjects. Logistic regression analyses were performed to identify the variables associated with multiple confounding factors. Statistical significance of the results was defined by *p* < 0.001 for most of the analyses, but by *p* < 0.05 for the logistic analysis with randomly selected small sample size.

## 3. Results

### 3.1. Participants’ Characteristics

The participants (*n* = 4490) consisted of 73.3% males and 26.7% females (Table 1). The average age of the participants was 47.4 ± 0.1 years with an average BMI of 22.69 ± 0.05. Of the participants, 15.5% were part-time or full-time night shift workers.

### 3.2. Correlation Analysis Focusing on Irregular Meal Timing

In the current analysis, we focused on the factor of irregular meal timing. Using the seven-step selection questionnaire (1 = strongly disagree to 7 = strongly agree), the questions determined whether the timing of daily food intake by the participants was regular or irregular, as previously described [32]. The participants who declared habits of irregular meal timing were most often night shift workers and of a young age, and they showed lower SWLS wellbeing scores (Table 1). To understand the effects of irregular meal timing, Spearman’s correlation coefficient between the scores of irregularity of meal timing and other questionnaires was evaluated (Figure 1). It is interesting to note that irregular meal timing was found to be positively associated with neuroticism and negatively associated with conscientiousness. Occupational presenteeism evaluated by the WHO-HPQ was also negatively associated with irregular meal timing. While absenteeism was not correlated with irregular meal timing, longer overtime hours did show a correlation with it. The participants with irregular meal timing also reported more infrequent physical activity and a dislike of exercising.

Irregular meal timing was also correlated with the evening chronotype and higher levels of social jet lag. Sleep quantity and quality on both workdays and free days were negatively correlated with irregular meal timing. Additionally, the scores of all factors in subjects’ sleep problems (referred to as category 5 of PSQI-J) [29] were higher in the participants with irregular meal timing. All subjective health and subjective mental health factors were also correlated with irregular meal timing. It is important to note that mental health was found to be more strongly correlated than subjective health with irregular meal timing. Although the stress level assessed was different in each questionnaire, all of these stress levels were more strongly correlated with irregular meal timing than the other factors shown in Figure 1.

### 3.3. Eating Habits with Irregular Meal Timing

To further understand the characteristics of irregular meal timing, another questionnaire on eating habits was used in this study (Figure 2). Irregular meal timing was strongly associated with a lower number of chewing cycles, shorter eating duration in each meal, eating a big meal at each mealtime, eating out frequently, eating an unbalanced diet frequently, eating salty food frequently, and low vegetable intake. Furthermore, irregular meal timing was associated with a lower frequency of breakfast consumption, higher nighttime snacking frequency, shorter period between last meal and sleep onset, lower ratio of breakfast, and higher ratio of snacks. These correlations were confirmed by the Kruskal–Wallis test (*p* < 0.001).

### 3.4. Logistic Regression Analysis for Subjective Health Outcomes

The strong association between irregular meal timing and subjective health outcomes was further analyzed by logistic regression analysis with anticipated confounding factors (Table 2). Each health outcome was set as an objective variable, and the score of irregularity of meal timing was set as the explanatory variable. Because of the seven-step answer of health outcomes, we divided the objective variable into two groups: answers 1 to 4 defined as “0”; answers 5 to 7 defined as “1”. Age, sex, BMI, night shift work, and SWLS score, total score of sleep problems, and the total daily physical activity level were set as confounding variables. Since the sample size was too large in the current data, false positive or negative errors could have occurred. Thus, we conducted the same analysis with randomly selected samples (*n* = 1405, Table 2). Under these adjustments, it was observed that the irregularity of meal timing was still significantly associated with most of the mental health outcomes with a higher correlation coefficient value and efficient odds ratio, but it was not significantly associated with most of the subjective health outcomes. Thus, irregularity of meal timing was found to be directly associated with subjective mental health outcomes.

## 4. Discussion

This study was a cross-sectional study with a large number of subjects (*n* = 4490) who were Japanese workers aged 20–69 years. This study described, for the first time, the association between irregular eating timing and other factors, including personal characteristics, working conditions, daily habits, and health comprehensively. Although there have been many studies discussing the timing of food intake, the irregularity of mealtimes was not the main focus of these studies [33]. For instance, breakfast skipping and nighttime snacking have been found to be associated with evening chronotype with greater social jet lag, which has also been associated with obesity or lifestyle-related diseases [11,12,13,20]. Breakfast skipping has also been confirmed to be associated with a higher incidence of obesity and hyperlipidemia [34]. In addition, it has been shown that delayed mealtimes or larger calorie intake at dinner leads to obesity with higher postprandial glucose levels [35]. Hedonic appetite is also higher in the evening, and it is controlled by the circadian clock in the dopaminergic neurons of the ventral tegmental area in mice [36]. “Eating jet lag”, defined by the difference in mealtimes between workdays and free days, was recently proposed, and it was found to be related to higher BMI independent of social jet lag and chronotype in young Mexican people aged 18–25 years [37]. Delayed lunch timing (approximately 3 pm) has also been reported to be related to obesity in Spain [38]. Therefore, timing and contents of food intake could be influenced by chronotype and sleep, and in turn which influence the regularity of meal timing [39,40,41,42,43]. However, the current study focused on the irregularity of daily mealtimes. Although we did not know the details of irregularity in each mealtime in the individuals, we found that the answer to one question about irregularity of meal timing was associated with a lower breakfast frequency, higher nighttime snacking, and unhealthy eating habits (e.g., salty, unbalanced, and low vegetable diets). Additionally, irregular meal timing was found to be independently associated with more health problems. Thus, this study provided important novel evidence to understand the association between daily eating habits and health. Consequently, future experiments should investigate irregularity in the timing of each daily meal using qualitative scales (e.g., variation of daily meal timing from 30 min to 2 h).

After comprehensively analyzing the background of the subjects, the resulting data suggested that irregular meal timing was associated with working conditions and personalities. The data showed an association between irregularity of meal timing, overtime work, and night shift work. The participants with these habits were primarily younger, neurotic, and not conscientious, with lower wellbeing scores. These characteristics may lead to a habit of irregular meal timing. A previous study reported that people who are neurotic have the following characteristics: greater overtime work, higher stress scores, and higher productivity losses [3]. People with low incomes may work overtime to supplement their income and consequently have sleep problems [44]. A previous study reported that irregular meal timing was associated with higher productivity loss through greater problems with sleep and stress [32]. Irregular meal timing was also found to be associated with the risk of cardiovascular diseases [45]. Additionally, night shift workers demonstrated the behaviors of irregular meal timing and unhealthy eating habits [46,47]. It was observed that social jet lag increased irregular meal timing and affected the quality and quantity of food intake [48]. Subjective health problems, including obesity, blood glucose, cholesterol, and blood pressure, were associated with irregular meal timing in the current study. These problems may lead to changes in leptin/ghrelin function, which might, in turn, lead to changes in mental health through metabolism [49]. Although many factors discussed in this study were related to each other, the investigation focused on the independent association between mealtime irregularity and health outcomes.

Because of the cross-sectional nature of the study, causal relationships between the variables were not investigated. Other limitations of this study were the unbalanced sex ratio (73.3% males) and race specificity (Japanese cohort). Data collected were solely self-reported, and the questionnaire (for daily eating habits) was not previously validated for the investigated population and, thus, may be subject to bias. Additionally, the high exclusion rate of the participants (4490 of 8720 participants were used for the analysis) may have decreased the generalizability of the current data. This study was performed during the worldwide COVID-19 pandemic, which may have affected the daily behavior and health of the participants [50]. However, a state of emergency was not announced by the government and the country was not under lockdown conditions during the current survey. In a previous study, we reported the effects of mild lockdown in Japan (from April 2020 to May 2020) on sleep and daily habits [51]. Compared with the data in our previous study, we do not think that the current data include the acute effect of the COVID-19 pandemic on irregular meal timing. Further longitudinal or interventional studies are necessary to verify the causality of irregular meal timing. Additionally, the effect of regular meal timing interventions must be evaluated in the future. The practice of TRE, in which people eat food during a period of 8–12 h per day, is a good example of such an intervention. TRE may regulate meal timing. It has been shown that TRE improved obesity and pre-type 2 diabetes by improving sleep and wellbeing [21,22]. Although TRE is an effective protocol for generating healthy eating habits, irregularities, such as lunch and afternoon snacks, are still under evaluation in TRE treatment. Finally, regular meal timing interventions may have a significant effect on health. For implementing health management in workplaces, an integrated health intervention system should be developed with scientific evidence, including the results of this study [4].

## Figures and Tables

**Figure 1 nutrients-13-02775-f001:**
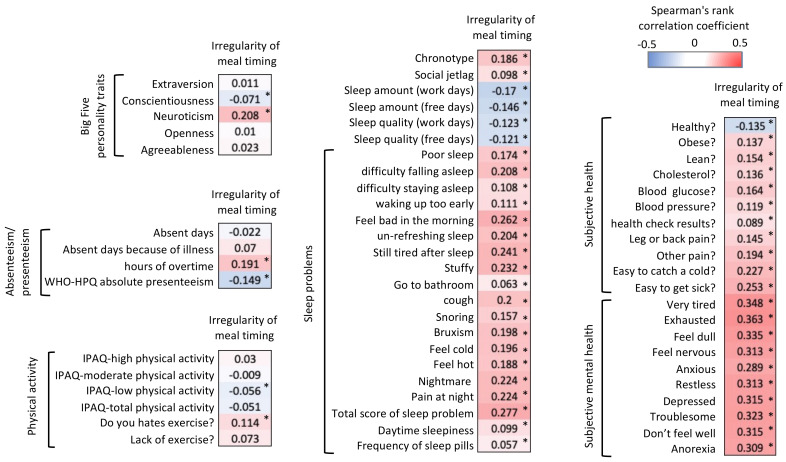
Spearman’s correlation analysis of “irregularity of meal timing” and other factors, indicated by the correlation coefficient values with gradient color in each column. Asterisks (*) in each column also indicate the significance of correlation (*p* < 0.001).

**Figure 2 nutrients-13-02775-f002:**
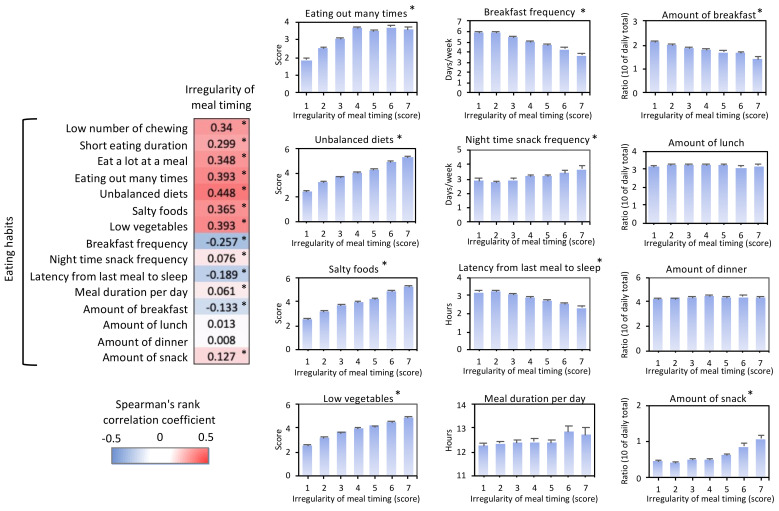
Irregular meal timing versus eating habits. Spearman’s correlation analysis (left panel) and averaged values (right panels) of “irregularity of meal timing” and other eating habits. Asterisks (*) in each column indicate the significance of correlation (*p* < 0.001) at the correlation analysis. The seven-step answer of each questionnaire was averaged to show in the right graph. Higher “score” means the result is similar to the “strongly agree” response for the questionnaire, as described in the methods section. Asterisks (*) on the upper right side in each graph also indicate the significance by the Kruskal–Wallis test (*p* < 0.001).

**Table 1 nutrients-13-02775-t001:** Participants’ characteristics.

			Regular Meal Timing	Irregular Meal Timing	*p* Value
Category		All (*n* = 4490)	(Score: 0–4; *n* = 3410)	(Score: 5–7; *n* = 1080)	Regular vs. Irregular
Sex	male % (*n*)	73.3 (3291)	73.5 (2507)	72.5 (783)	0.497
	female % (*n*)	26.7 (1199)	26.5 (904)	27.5 (297)	
Age	(years)	47.41 ± 0.17	48.06 ± 0.20	45.36 ± 0.34	< 0.001
BMI	(kg/m^2^)	22.69 ± 0.05	22.64 ± 0.06	22.83 ± 0.12	0.138
Night shift worker	% (*n*)	15.5 (695)	11.4 (388)	28.4 (307)	< 0.001
SWLS	(score)	17.24 ± 0.09	17.52 ± 0.10	16.35 ± 0.20	< 0.001

Data expressed as mean ± SEM. SWLS: Satisfaction with Life Scale.

**Table 2 nutrients-13-02775-t002:** Logistic regression analysis for subjective health outcomes.

	All Samples (*n* = 4490)				Randomly Selected Samples (*n* = 1405)			
Objective Variable	Irregularity of Meal Timing				Irregularity of Meal Timing			
	B	*p*	OR	95% CI	B	*p*	OR	95% CI
Subjective health								
Healthy?	−0.111	*p* < 0.001	0.895	0.856, 0.936	−0.107	*p* < 0.01	0.898	0.828, 0.974
Worried about being obese?	0.079	*p* < 0.01	1.082	1.032, 1.134	0.094	*p* < 0.05	1.098	1.007, 1.197
Worried about being lean?	0.163	*p* < 0.001	1.177	1.101, 1.258	0.111	0.08	1.117	0.987, 1.264
Worried about your blood cholesterol level?	0.063	*p* < 0.01	1.065	1.018, 1.113	0.021	0.618	1.021	0.941, 1.108
Worried about your blood glucose level?	0.072	*p* < 0.01	1.074	1.022, 1.129	0.07	0.136	1.072	0.978, 1.175
Worried about your blood pressure level?	0.048	*p* < 0.05	1.049	1.000, 1.101	0.048	0.284	1.049	0.961, 1.146
Worried about your health check results?	0.022	0.117	1.034	0.992, 1.079	−0.039	0.33	0.962	0.889, 1.04
Having pain in your leg or back?	0.039	0.075	1.04	0.996, 1.086	0.014	0.73	1.014	0.937, 1.098
Having pain in your other place?	0.109	*p* < 0.001	1.115	1.068, 1.164	0.072	0.078	1.074	0.992, 1.163
Easy to catch a cold?	0.14	*p* < 0.001	1.15	1.085, 1.218	0.114	*p* < 0.05	1.121	1.001, 1.255
Easy to get sick?	0.171	*p* < 0.001	1.187	1.111, 1.268	0.111	0.091	1.117	0.983, 1.271
Subjective mental health								
Very tired	0.232	*p* < 0.001	1.262	1.206, 1.320	0.215	*p* < 0.001	1.239	1.142, 1.346
Exhausted	0.269	*p* < 0.001	1.309	1.246, 1.375	0.304	*p* < 0.001	1.355	1.236, 1.485
Feel dull	0.219	*p* < 0.001	1.245	1.187, 1.305	0.246	*p* < 0.001	1.279	1.172, 1.395
Feel nervous	0.214	*p* < 0.001	1.239	1.182, 1.298	0.311	*p* < 0.01	1.365	1.110, 1.679
Anxious	0.152	*p* < 0.001	1.164	1.111, 1.220	0.243	*p* < 0.05	1.275	1.044, 1.557
Restless	0.185	*p* < 0.001	1.203	1.142, 1.268	0.208	0.054	1.231	0.996, 1.521
Depressed	0.187	*p* < 0.001	1.205	1.147, 1.267	0.283	*p* < 0.01	1.327	1.076, 1.635
Troublesome	0.201	*p* < 0.001	1.223	1.165, 1.283	0.526	*p* < 0.001	1.691	1.345, 2.128
Don’t feel well	0.205	*p* < 0.001	1.227	1.169, 1.289	0.355	*p* < 0.01	1.426	1.158, 1.757
Anorexia	0.36	*p* < 0.001	1.433	1.319, 1.557	0.33	*p* < 0.05	1.391	1.006, 1.925

We divided the objective variable into 2 groups: answers 1 to 4 defined as “0”; answers 5 to 7 defined as “1”. For each objective variable, the coefficient (B), *p* value, odds ratio (OR), and 95% confidence intervals (CI) were indicated. Confounding factors: age, sex, BMI, night shift work, SWLS, sleep problem score, and IPAQ total daily physical activity.

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
