# Peer review of "Association between Irregular Meal Timing and the Mental Health of Japanese Workers"

_nutrients, 2021, doi:10.3390/nu13082775_

Round 1
Reviewer 1 Report
Comments on Nutrients-1320514 “The impact of irregular meal timings on the mental health of 2 Japanese workers”
Using web-based survey, the authors examined the association of irregular meal timing with certain health conditions among 4490 Japanese workers and conclude that irregular meal timing is a good marker of health outcomes. Below are reviewer’s comments for the authors’ consideration:
- This is a cross-sectional study. Theoretically, the cause relationship cannot be established, but the title might suggest a cause relation could be determined. removing “impact” from the title may help to avoid the confusion. in addition, the key variable irregular meal timings need more description, for instance, how many items were in the survey used to determine it? what were they? was it an index from a sum of scores of this or these item(s)? what was its range and what does the score mean? what was the cut-off used to define the irregular meal timings and why? Was this concept from previous studies or proposed by authors? If it was from previous studies, please provide reference citations, otherwise, authors need to inform readers whether there was any pilot or similar study being conducted before this one in regard of its countability and/or reliability. Without such information, it would be difficulty for readers to agree with your conclusion that it is a good marker of health outcomes.
- Based on the title, mental health should be another key variable in this study, but unfortunately, there was no clear description of it. It seems the authors try to explore every aspect of health including obesity, physical activity, sleep issues, personality, etc. although directly or indirectly all of them may somehow be associated with the two key variables or the main theme listed in the title, however, as researchers, we need to sort these unorganized, mixed, seemed to be chaos puzzles in a way to show a logic picture. I have to say, the information collected is rich, but the study presented here lost its focus.
- A total 8720 workers participated the survey, and only 4490 of them (51.5%) were included in this study; nearly half of workers were excluded from the analysis due to not completing the survey. It is a normal phenomenon to have incomplete surveys, but one needs to conduct that an attrition analysis to compare those included vs those excluded on certain main characteristics, eg, age, sex, night shift worker to make sure whether the results from this study can generalized to the population where the sample was from. The results of this comparison should be discussed in limitation.
- Since the research topic was related to timings, in the description of the survey, I suggest adding information of an introduction to tell readers that how many items were together included in this survey? how long does it take to complete on average? Was there any minimum literacy requirement for understanding questions asked? What was other demographic information was collected besides age, gender, such as education and marriage? which might be potential confounding factors as well. In the suggested attrition analysis, please pay attention to whether those night shift workers were more unlikely to complete the survey.
- The issues in statistical analysis: a) type I errors in multiple comparisons would be expected with such larger number of tests on irregular meal timing (BTW, I noted that two forms were used exchangeable, ie, “irregular meal timings” vs “irregular meal timing”. Was there any difference between them? If yes, please specify); b) one-way ANOVA and multivariate linear regression both need to check whether assumptions are met, otherwise, the calculated test statistic is not countable. Not sure if this was done; c) interpretation of Spearman rank correlations should follow the common rules, eg, in general, the correlations are considered as weak if r(s) <0.4, moderate 0.4 – 0.7, and strong or high >=0.7. the rank correlation coefficients in figure 1 were all less than 0.4, but “all of these stress levels were highly correlated with irregular meal timing.” Seems misleading; d) since sample size was large, any smaller difference would be easier detected, p-values only reflect the likelihood of detecting these differences, but not telling the actual effect. Therefore, I suggest authors considering use effect size to address the potential impact.
- In discussion, a) it should match with the theme presented in title; b) “Since the current study comprehensively analyzed the background of the subjects, their data suggest that the cause of irregular meal timing may be attributed to their working conditions and personalities.” Seems overstating the cause relationship; c) for the sex issue, please provide what would be a general estimation of sex ratio in Japanese working force, and discuss how this differ from the population, particularly, whether sex ratio from those excluded was similar to the results presented here; d) please addressing the issues listed previously.
- Other issues: a) in table 1, please add “mean, SD” for continuous variables, and add (n) for categorical variables; also please make a footnote spelling out SWLS; b) it is difficult to follow what the authors presented those bar charts in figure 2. There seemed no title to figure 2; they were to me footnotes, in which I have a question on “high score means the result is similar to …” how high score was defined? Were they the same for different items?
Author Response
Thank you for your review and kind comments.
Using web-based survey, the authors examined the association of irregular meal timing with certain health conditions among 4490 Japanese workers and conclude that irregular meal timing is a good marker of health outcomes. Below are reviewer’s comments for the authors’ consideration:
We appreciate your review and kind comments. We have addressed your suggestions carefully in the revised manuscript, and indicated the responses below in the red.
- This is a cross-sectional study. Theoretically, the cause relationship cannot be established, but the title might suggest a cause relation could be determined. removing “impact” from the title may help to avoid the confusion. in addition, the key variable irregular meal timings need more description, for instance, how many items were in the survey used to determine it? what were they? was it an index from a sum of scores of this or these item(s)? what was its range and what does the score mean? what was the cut-off used to define the irregular meal timings and why? Was this concept from previous studies or proposed by authors? If it was from previous studies, please provide reference citations, otherwise, authors need to inform readers whether there was any pilot or similar study being conducted before this one in regard of its countability and/or reliability. Without such information, it would be difficulty for readers to agree with your conclusion that it is a good marker of health outcomes.
Thank you for your kind suggestions. We completely agree that our study cannot investigate any causality. We have, therefore, changed the title to “Association between irregular meal timing and the mental health of Japanese workers.”
There was only one question used for defining irregular meal timing, with the 7-step answer scale. A similar question was used in a previous study wherein the association between irregular meal timing and presenteeism was investigated (Hayashida et al., Neuropsychiatr Dis Treat, 2021). We added more explanation in the methods section, as below.
“For example, the question for the irregularity of meal timing was “Think about your eating behavior in the past 4 weeks: Eat timing is irregular”, followed by the seven-step answer scale as described above.”
Also in the results section, we added the reference, “In the current analysis, we focused on the factor of irregular meal timing. Using the seven-step selection questionnaire (1= strongly disagree to 7= strongly agree), the questions determined whether the timing of daily food intake of the participants was regular or irregular, as previously described (Hayashida et al., 2021)”
- Based on the title, mental health should be another key variable in this study, but unfortunately, there was no clear description of it. It seems the authors try to explore every aspect of health including obesity, physical activity, sleep issues, personality, etc. although directly or indirectly all of them may somehow be associated with the two key variables or the main theme listed in the title, however, as researchers, we need to sort these unorganized, mixed, seemed to be chaos puzzles in a way to show a logic picture. I have to say, the information collected is rich, but the study presented here lost its focus.
Thank you for your suggestions. We added more explanations on the mental health questions in the methods section, as below.
“The question used was “Think about your mental health condition: I feel very tired”, along with the seven-step selection answer scale as described above.”
As already described in the manuscript, we chose those questionnaires based on the Japanese Stress Check Program, which is currently authorized by the Japanese Ministry of Health, Labor and Welfare and is considered a standard measure for the assessment of occupational stress (Kawakami et al., 2016; Moriguchi et al., 2020).
Although the current data is rich, we focused on the daily habits and health outcomes in this study. First, we tried to find factors correlated with irregular meal timing and found that health outcomes were more correlated with it. Second, we tried to understand more about irregular meal timing by comparing it with the other eating habits. Third, we tried to confirm the correlation between the irregular meal timing and health outcomes under the confounding variables, such as age, sex, sleep, night-shift work, or physical activity. In the future study, we would like to analyze further for other factors with questions that were not used in the current analysis. Thus, the current analysis focused on irregular meal timing only.
- A total 8720 workers participated the survey, and only 4490 of them (51.5%) were included in this study; nearly half of workers were excluded from the analysis due to not completing the survey. It is a normal phenomenon to have incomplete surveys, but one needs to conduct that an attrition analysis to compare those included vs those excluded on certain main characteristics, eg, age, sex, night shift worker to make sure whether the results from this study can generalized to the population where the sample was from. The results of this comparison should be discussed in limitation.
We apologize for the inadequate explanation of the survey protocol. Since the survey was conducted using an online web system, the participants who were not working at the time, who were under 20 years old, older than 69 years old or did not complete the questionnaire appropriately, were automatically excluded. Thus, out of total 8720 participants, 4490 participants finally completed the questionnaire. Therefore, the attrition analysis cannot be done because we do not have detailed information of the participants who were excluded in the current analysis.
- Since the research topic was related to timings, in the description of the survey, I suggest adding information of an introduction to tell readers that how many items were together included in this survey? how long does it take to complete on average? Was there any minimum literacy requirement for understanding questions asked? What was other demographic information was collected besides age, gender, such as education and marriage? which might be potential confounding factors as well. In the suggested attrition analysis, please pay attention to whether those night shift workers were more unlikely to complete the survey.
The questionnaire includes 276-318 questions based on the working style (e.g., shift work or not). It includes detailed questions related to demographics, working conditions, COVID-19 etc. However, we focused on the irregular meal timing for the current analysis and the other questions would be used for analysis in the future. Although we agreed that marriage, education history, housemate, income, etc. might be confounding variables for the health outcome, we added the main demographics, daily habits, and SWLS (Satisfaction with Life Scale) for confounding in the regression analysis. SWLS may reflect other demographic factors.
The participants answered the questions within 30-50 min. We do not know whether shift workers took more time to answer or not.
We have added information about the survey in the methods section.
“The questionnaire includes 276-318 questions depends on the working style (e.g., shift work or not); the participants answered them within 30-50 min. In the current study, we focused on the below described variables from the elaborate questionnaire.”
- The issues in statistical analysis: a) type I errors in multiple comparisons would be expected with such larger number of tests on irregular meal timing (BTW, I noted that two forms were used exchangeable, ie, “irregular meal timings” vs “irregular meal timing”. Was there any difference between them? If yes, please specify); b) one-way ANOVA and multivariate linear regression both need to check whether assumptions are met, otherwise, the calculated test statistic is not countable. Not sure if this was done; c) interpretation of Spearman rank correlations should follow the common rules, eg, in general, the correlations are considered as weak if r(s) <0.4, moderate 0.4 – 0.7, and strong or high >=0.7. the rank correlation coefficients in figure 1 were all less than 0.4, but “all of these stress levels were highly correlated with irregular meal timing.” Seems misleading; d) since sample size was large, any smaller difference would be easier detected, p-values only reflect the likelihood of detecting these differences, but not telling the actual effect. Therefore, I suggest authors considering use effect size to address the potential impact.
Thank you for your suggestions for the statistical analysis. Based on your suggestions, we collected our analysis indicated below.
- We used “irregular meal timing” in the revised version.
- Since the one-way ANOVA was not appropriate in Figure 2, we used non-parametric analysis (Kruskal-Wallis test) in the revised manuscript.
- We agreed that the current sample size is too large. Accordingly, the power value was 1.0 for the current multivariate regression analysis (calculated by G*Power 3.1). To avoid the false negative or positive errors in the multivariate regression analysis, we performed power analysis (minimum effect size, 0.02; p-value, 0.001; power, 0.8; predictors, 8) and found the appropriate sample size of N = 1405. Thus, we used the randomly selected 1405 samples and performed the multivariate linear regression analysis (with the same confounding of model 2 in Table 2). The results were indicated below in Table S1 (only for reviewers). We found the results similar to those in Table 2. We indicated those findings in the results section in the revised manuscript, as below.
“Since the sample size was too large in the current data, false positive or negative errors could have occurred. However, we confirmed the same results with randomly selected samples (N = 1405).”
- For the correlation analysis in Fig.1, we agreed that the correlation r(s) < 0.4 is not high. Thus, we changed the sentence in the results section to “Although the stress level assessed in each questionnaire was different, all of these stress levels were more correlated with irregular meal timing than other factors shown in Fig. 1.”
- The effect size for the correlation analysis was already indicated by r. and for the multivariate regression analysis it was indicated by R2. We chose p < 0.001 as significance for all analysis. In Fig. 1, we agreed some correlations were not enough to explain because of the small r. But we focused on the subjective health outcomes which had enough correlations with irregular meal timing. Additionally, based on the above analysis (multivariate regression analysis with randomly selected samples) we believe our current conclusion is true.
Table S1 for the reviewers only. Multivariate linear regression analysis for subjective health outcomes (N = 1405). For each objective variable, the standardized coefficient (β) was indicated with a p value. R-squared value was used to assess the fitness of each model. Confounding factors for model 2: age, sex, BMI, night shift work, SWLS, sleep problem score, and IPAQ total daily physical activity.
- In discussion, a) it should match with the theme presented in title; b) “Since the current study comprehensively analyzed the background of the subjects, their data suggest that the cause of irregular meal timing may be attributed to their working conditions and personalities.” Seems overstating the cause relationship; c) for the sex issue, please provide what would be a general estimation of sex ratio in Japanese working force, and discuss how this differ from the population, particularly, whether sex ratio from those excluded was similar to the results presented here; d) please addressing the issues listed previously.
Thank you for your suggestions.
We have added in more discussion about the relationship between mental health and irregular meal timing.
“Subjective health problems of obesity, blood glucose, cholesterol, and blood pressure were associated with irregular meal timing in this study. These problems may lead to changes in leptin/ghrelin function, which might in turn lead to changes in mental health through metabolism (Horne et al., 2018).”
We changed the sentence to “Since the current study comprehensively analyzed the background of the subjects, their data suggest that the irregular meal timing was associated with their working conditions and personalities.”
For the sex ratio, we confirmed that it was similar to the sex ratio of Japanese full-time workers reported in the national survey from Statistics Bureau. Actually, population size (over 15 years old) is N = 23,340,000 for male workers and N = 12,000,000 for female workers in Japan. We added this information in the methods section as below.
“The sex ratio of the participants was similar to the sex ratio of Japanese full-time workers as reported in the Japanese national survey from Statistics Bureau.”
- Other issues: a) in table 1, please add “mean, SD” for continuous variables, and add (n) for categorical variables; also please make a footnote spelling out SWLS; b) it is difficult to follow what the authors presented those bar charts in figure 2. There seemed no title to figure 2; they were to me footnotes, in which I have a question on “high score means the result is similar to …” how high score was defined? Were they the same for different items?
Thank you for your suggestions. We corrected the Table 1 as you suggested.
In addition, we revised the figure legend of Fig. 2. The high score was defined as per the 7-step answer scale of the questionnaires (from 1 to 7). We added more explanation on this.
The revised sentences are “Spearman’s correlation analysis (left panel) and averaged values (right panels) between “irregularity of meal timing” and other eating habits. Asterisks (*) in each column indicate the significance of correlation (p < 0.001) at the correlation analysis. The seven-step answer of each questionnaire was averaged to show in the right graph. Higher “score” means the result is similar to the “strongly agree” response for the questionnaire, as described in the methods section. Asterisks (*) on the upper right side in each graph also indicate the significance by the Kruskal-Wallis test (p < 0.001).”

Reviewer 2 Report
Authors explore the effect of meal timing habits toward mental health features in Japanese workers. The article is novel, especially given the rising prevalence of both mental disorders and obesity. However, I have some suggestions mainly regarding introduction and discussion section.
Comments:
- Line 35 authors should provide a comprehensive reference for the statement i.e. PMID: 33549913.
- Line 39 these food groups contain also polyphenols which have been demonstrated to exert beneficial effects toward mental health through various pathways (direct by passing blood-brain barriers, and indirect by modulation of gut microbiota and systemic inflammation/oxidatie stress please see: PMID: 32340112.
- Authors explored among outcomes metabolic features and so authors should consider discussing ghrelin leptin cycle as possible mechanism mediating the effect of food toward mental health through metabolism.
- Line 180 “Although there 180 have been many studies discussing the timing of food intake,” I believe authors should provide a comprehensive reference for this statement i.e. PMID: 33356688.
- Authors should consider the following as another limitation of the study: data are solely self-reported and the questionnaire (for daily eating habits) was not previously validated for the investigated population and thus may be subjected to bias.
Author Response
Authors explore the effect of meal timing habits toward mental health features in Japanese workers. The article is novel, especially given the rising prevalence of both mental disorders and obesity. However, I have some suggestions mainly regarding introduction and discussion section.
We appreciate your review and kind comments. We have addressed your suggestions carefully in the revised manuscript, and indicated the responses below in the red.
Comments:
- Line 35 authors should provide a comprehensive reference for the statement i.e. PMID: 33549913.
Thank you for your suggestion. We have added the reference in the revision.
- Line 39 these food groups contain also polyphenols which have been demonstrated to exert beneficial effects toward mental health through various pathways (direct by passing blood-brain barriers, and indirect by modulation of gut microbiota and systemic inflammation/oxidatie stress please see: PMID: 32340112.
Thank you for your suggestion. We have added this sentence with the reference. “Additionally, polyphenol compounds are good for mental health; they modulate brain function, gut microbiota, and oxidative stress/inflammation signaling (Godos et al., 2020)”.
- Authors explored among outcomes metabolic features and so authors should consider discussing ghrelin leptin cycle as possible mechanism mediating the effect of food toward mental health through metabolism.
Thank you for your suggestion. We have added discussion in the revision, as below.
“Subjective health problems of obesity, blood glucose, cholesterol, and blood pressure were associated with irregular meal timing in this study. These problems may lead to changes in leptin/ghrelin function, which might in turn lead to changes in mental health through metabolism (Horne, 2018).”
- Line 180 “Although there 180 have been many studies discussing the timing of food intake,” I believe authors should provide a comprehensive reference for this statement i.e. PMID: 33356688.
Thank you for your suggestion. We have added the reference “Currenti et al., 2020” in the discussion.
- Authors should consider the following as another limitation of the study: data are solely self-reported and the questionnaire (for daily eating habits) was not previously validated for the investigated population and thus may be subjected to bias.
Thank you for your suggestion. We have added this limitation in the discussion section.

Round 2
Reviewer 1 Report
Comments on Nutrients-1320514 revised submission
Thanks to the authors’ efforts to improve the quality of the manuscript. There are still some concerns for authors’ consideration:
- Although the new title “association between irregular meal timing and the mental health of Japanese workers” reflects the cross-sectional study design, it still did not match with what was given in context. Since mental health was only a smaller portion of the outcomes, it would be less confused if the “mental health” in the title to be replaced with another term that is reflecting what the paper talks about, for instance “subjective health outcomes”.
- Please use the response part “Since the survey was conducted using an online web system, the participants who were not working at the time, who were under 20 years old, older than 69 years old or did not complete the questionnaire appropriately, were automatically excluded. Thus, out of total 8720 participants, 4490 participants finally completed the questionnaire.” to describe the process of sample selection, which is clearer than what was currently showed in the context. BTW, I guess, the 8720 people were those who were eligible to participate the survey, but 4490 were actually available to participate the survey. I would suggest having a discussion of its generalizibility.
- Please add the citation for sex-ratio, ie, the report of Japanese national survey from statistics bureau.
- Should “These correlations were confirmed by one-way ANOVA (p<.001) (pg5, line 172)” be updated since non-parametric methods were used?
- The change in title of figure 2 does not satisfy what has been requested. A title should be outline what would be described in the figure even without looking at figure itself. Here is a suggested title for authors’ consideration – “irregular meal timing versus eating habits” and what have been used in the current title can be treated as footnotes. For figure 1, the title can be modified as “Spearman’s correlation analysis between “irregularity of meal timing” and other factors.” Treat the rest of them as footnotes.
6. For multivariate linear regression analyses, the same problems as did one-way ANOVA previously, a) normality for dependent variables since they were all results from the 7-step questionnaire (at least no checking assumption was done and reported here); b) type I error for multiple comparison. The authors tried to use a random sample (n = 1405) to repeat the analyses and there is no doubt to have a similar result, but it does not overcome the problem of increasing type I error due to multiple comparison. A potential solution might be to combine those highly correlated variables into an index by summary of their scores (factor analysis may help) to increase the variability and improve its normality distribution, eg, health worry index (five of them, each range from 1 – 7, the index range may vary from 5 to 35). BTW, please change “health outcomes” in the 1st sentence under section 3.4 multivariate …. To “subjective health outcomes” to match with what described in table 2 title since they are two different concepts.
Author Response
We uploaded our response letter here.
